# Evaluating the Properties of Ginger Protease-Degraded Collagen Hydrolysate and Identifying the Cleavage Site of Ginger Protease by Using an Integrated Strategy and LC-MS Technology

**DOI:** 10.3390/molecules27155001

**Published:** 2022-08-06

**Authors:** Wei Liu, Wenning Yang, Xueyan Li, Dongying Qi, Hongjiao Chen, Huining Liu, Shuang Yu, Guopeng Wang, Yang Liu

**Affiliations:** 1Department of Chemistry of Traditional Chinese Medicine, School of Chinese Materia Medica, Beijing University of Chinese Medicine, Beijing 100102, China; 2Zhongcai Health (Beijing) Biological Technology Development Co., Ltd., Beijing 101500, China

**Keywords:** ginger protease, bioactive peptides, bioavailability, peptides identification

## Abstract

(1) Methods: An integrated strategy, including in vitro study (degree of hydrolysis (DH) and 2,2-diphenyl-1-picrylhydrazyl (DPPH) radical scavenging activity) and in vivo study (absorption after oral administration in rats), was developed to evaluate the properties of the fish skin gelatin hydrolysates prepared using different proteases (pepsin, alkaline protease, bromelain, and ginger protease). Meanwhile, in order to identify the hydrolysis site of ginger protease, the peptides in the ginger protease-degraded collagen hydrolysate (GDCH) were comprehensively characterized by liquid chromatography/tandem mass spectrometry (LC-MS) method. (2) Results: The GDCH exhibited the highest DH (20.37%) and DPPH radical scavenging activity (77.73%), and in vivo experiments showed that the GDCH was more efficiently absorbed by the gastrointestinal tract. Further oral administration experiments revealed that GDCH was not entirely degraded to free amino acids and can be partially absorbed as dipeptides and tripeptides in intact forms, including Pro-Hyp, Gly-Pro-Hyp, and X-Hyp-Gly tripeptides. LC-MS results determined the unique substrate specificity of ginger protease recognizing Pro and Hyp at the P_2_ position based on the amino acids at the P_2_ position from the three types of tripeptides (Gly-Pro-Y, X-Hyp-Gly, and Z-Pro-Gly) and 136 identified peptides (>4 amino acids). Interestingly, it suggested that ginger protease can also recognize Ala in the P_2_ position. (3) Conclusions: This study comprehensively evaluated the properties of GDCH by combining in vitro and in vivo strategies, and is the first to identify the cleavage site of ginger protease by LC-MS technique. It provides support for the follow-up study on the commercial applications of ginger protease and bioactivities of the hydrolysate produced by ginger protease.

## 1. Introduction

Ginger protease (also known as Zingibain, EC3.4.22.67) is a plant protease from the ginger rhizome, *Zingiber Officinale Roscoe*, first reported by Thompson et al. [1]. It is considered as a green and safe food additive for industrial applications, including wine clarification, milk curdling [2], and meat tenderization [3]. Furthermore, many studies have shown that ginger protease-degraded collagen hydrolysate (GDCH) can be used as a functional food for patients with obesity [4], osteoporosis [5], gastrointestinal dysfunction [6], and type 2 diabetes [7].

In addition to ginger protease, other enzymes such as alkaline protease, bromelain, and pepsin are commonly used to hydrolyze gelatin and study the activity of their hydrolysates. Several studies have indicated that hydrolysates obtained using ginger protease exhibited higher degree of hydrolysis (DH) or antioxidative activity compared with those obtained using bromelain [8] and pepsin-pancreatin [9]. However, few studies have considered comparing the in vivo absorption of different hydrolysates after oral ingestion, which may be closely related to the actual effect of collagen peptides in vivo. Hyp, a characteristic amino acid of collagen, is often used to quantify the absorption and dynamic changes after oral intake of hydrolysates [10,11]. Furthermore, it is reasonable to evaluate the in vivo absorption by the dynamic changes of Hyp after oral administration as Hyp is one of the amino acids closest to the regression curve between the increased amount in plasma and its content in gelatin among all amino acids [12,13]. Therefore, a comprehensive analysis may be a better way to compare the efficiency of different enzymatic hydrolysates in vitro and in vivo.

Moreover, it has been shown that the cleavage sites of enzymes affect the amino acid sequences and antioxidative properties of the hydrolysates [14,15]. Therefore, it is of great significance to identify the cleavage site of ginger protease. In 1994, Yongjun Duan first revealed that ginger protease showed preference for cleavage at Pro peptide bonds by using dipeptide specificity mapping [16]. In 1999, this hydrolysis specificity was confirmed by a kinetic analysis of 20 tripeptide substrates as k_cat_/K_m_ values for substrates with P_2_ Pro were 100–3000 times greater than those with other amino acids at P_2_ [17]. Then, Kim et al. investigated the cleavage site of ginger protease to hydrolyze fluorescent proline-containing substrates, and the results also revealed that ginger protease preferentially cleaved peptide bonds with Pro at the P_2_ position. Recently, Taga et al. confirmed that ginger protease can recognize Hyp at the P_2_ position due to the substantial production of X-Hyp-Gly tripeptides [18]. However, in the above reports, only a few dipeptides or tripeptides were studied as substrates to speculate the hydrolysis site of ginger protease. Due to the lack of comprehensive identification of the peptides in the hydrolysates and therefore the insufficient understanding of the components in the samples, there is a possibility of false positive results. Liquid chromatography/tandem mass spectrometry (LC-MS) technology has been successfully applied in the rapid determination of cleavage site of numerous proteases, characterized by both high sensitivity and high selectivity [19,20]. Therefore, we intend to identify the peptides in GDCH by LC-MS, and then determine the hydrolysis site of ginger protease.

Based on the above analysis, the objectives of this study were to compare the properties of hydrolysate obtained from fish skin gelatin using ginger protease with those produced using pepsin, alkaline protease, and bromelain. In addition, the hydrolysis site of ginger protease was first identified based on the amino acids at the P_2_ position of peptides (>4 amino acids) in the GDCH by nano liquid chromatography coupled with electrospray ionization tandem mass spectrometry (nano LC-ESI-MS/MS). The flowchart of the study is presented in Figure 1.

## 2. Results

### 2.1. Comparison of the Hydrolysis Effect In Vitro

There were many factors affecting the functional properties and bioactivity of the hydrolysates, including protein species, hydrolysis sites of enzymes, and hydrolysis conditions [21]. In this study, pepsin, alkaline protease, bromelain, and ginger protease were used to hydrolyze gelatin. As shown in Figure 2, DH and 2,2-diphenyl-1-picrylhydrazyl (DPPH) radical scavenging activity were significantly affected by enzyme type (*p* < 0.05). The GDCH exhibited a significantly higher DH and in vitro antioxidant activity than the hydrolysates prepared using pepsin, alkaline protease, and bromelain (*p* < 0.05). As a whole, among the four types of hydrolysates, ginger protease appeared to be the most effective one in hydrolyzing gelatin.

### 2.2. Comparison of the Hydrolysis Effect via Estimation of Total Hyp in Rat Plasma In Vivo

The results presented in Figure 3 showed the changes over time in the total Hyp concentrations in rat plasma after ingestion of the gelatin and hydrolysates (HP, HA, HB, and GDCH). The results revealed that the total Hyp in plasma rapidly increased and reached a maximum level after 1–2 h. Then, the concentrations gradually decreased and reached negligible levels 8 h after ingestion.

The T_max_ of total Hyp after ingestion was 2.00 ± 0.00 h (gelatin), 2.00 ± 0.00 h (HP), 1.83 ± 0.41 h (HA), 1.83 ± 0.41 h (HB), and 1.67 ± 0.52 h (GDCH). Although no significant differences were observed, the T_max_ of the total Hyp after GDCH ingestion was advanced to an earlier time (1.67 ± 0.52 h). Moreover, the C_max_ and AUC_0–8 h_ of total Hyp of GDCH were significantly higher than those of gelatin, HP, HA, and HB (*p* < 0.05) (Table 1).

### 2.3. Molecular Weight Distribution of the GDCH

The molecular weight distribution of GDCH was analyzed by size exclusion chromatography (SEC). Ginger protease can hydrolyze fish skin gelatin to peptides with a low average molecular weight (938 Da). As shown in Table 2, GDCH had a large quantity (72.86%) of oligopeptides with molecular weight (MW) below 1000 Da, particularly 189–576 Da (42.19%). It is important to highlight that those dipeptides and tripeptides are mainly in the range of 189–576 Da, which can be directly absorbed and utilized by the human body via the peptide transport carriers.

### 2.4. Identification of Dipeptides and Tripeptides in the GDCH

Since ginger protease had a better hydrolysis effect, the dipeptides and tripeptides of the GDCH were evaluated in more detail. As shown in Table 3, the most abundant oligopeptides were Z-Pro-Gly tripeptides (Ala-Pro-Gly and Leu-Pro-Gly) and Gly-Pro-Y tripeptides, including Gly-Pro-Ala, Gly-Pro-Val, Gly-Pro-Gln, Gly-Pro-Glu, and Gly-Pro-Arg, suggesting that ginger protease can recognize Pro at the P_2_ position. Moreover, X-Hyp-Gly tripeptides (such as Leu/Ile-Hyp-Gly and Phe-Hyp-Gly) were also detected, which may be produced by the protease activity of ginger protease cleaving peptides towards Hyp at the P_2_ position. Furthermore, the results also revealed that Hyp-Gly and Gly-Pro-Hyp-Gly existed in large amounts.

### 2.5. Identification of Dipeptides and Tripeptides in Rat Plasma after GDCH Ingestion

The collagen peptides in plasma were further identified by LC-MS/MS. The Hyp-containing peptides, including Pro-Hyp, Gly-Pro-Hyp, and X-Hyp-Gly tripeptides (Pro-Hyp-Gly, Leu/Ile-Hyp-Gly, Phe-Hyp-Gly), were identified in plasma after GDCH ingestion (Table 4).

### 2.6. Peptide Identification and Sequencing by Nano LC-ESI-MS/MS

In this study, a total of 136 peptides were identified in the GDCH (Appendix A). The number of amino acids in these peptides ranged from 7 to 22, and the MW from 700 to 2300 Da. Most of the peptides were within 15 amino acids in length and had a molecular weight below 1500 Da. It has been reported that peptides with a molecular weight below 2000 Da can cross the intestinal barrier more easily [22].

The Peptide Ranker is based on a novel N-to-1 neural network for efficiently scoring and screening novel bioactive peptides. The 28 peptides showed high probability of biological activity predicted by Peptide Ranker (>0.8) (Table 5). Further research is needed to investigate the potential bioactivity of those screened peptides.

Notably, the amino acid residues at the P_2_ position in the above-mentioned peptides were Pro (66.91%) and Hyp (14.71%) as shown in Table 6, which was due to the activity of ginger protease preferentially cleaving peptide peptides towards Pro and Hyp at the P_2_ position. Additionally, the results showed that Ala accounted for a small part (9.56%), indicating ginger protease can recognize Ala at the P_2_ position.

## 3. Discussion

Commercial enzymes such as alkaline protease, bromelain, and pepsin are commonly used in daily life, while the commercial value of ginger protease has yet to be developed. In an attempt to illustrate the hydrolysis effects of different enzymes, a comparison between ginger protease and other proteases in hydrolyzing fish skin gelatin was performed in vitro and in vivo. In vitro studies showed that GDCH exhibited the highest DH and DPPH radical scavenging activity. The possible reasons for the excellent antioxidant activity of GDCH are as follows. First, GDCH has the highest DH, and previous studies have found that the DH can exert considerable impact on the antioxidative activity of the resulting hydrolysates [23,24]. Second, the proportion of peptides with a MW below 1000 Da in the GDCH was 72.86%, and a previous study showed that peptides within this mass range exhibited higher antioxidant activity [9]. Third, the antioxidative activity of GDCH might be partly due to the high content (42.19%) of dipeptides and tripeptides, which have been shown to play a crucial role in antioxidative activity [25]. Fourth, some peptides in GDCH (Table 3 and Appendix A) contain some characteristic amino acids, such as Leu, Hyp, Val, Gly, Pro, and Tyr, which can significantly increase the antioxidant activity of polypeptide sequences [26,27,28]. Based on the above analysis, GDCH is expected to be a promising natural antioxidant to prevent the oxidative process in vivo, and research has confirmed that GDCH can induce glutathione synthesis to protect against hydrogen peroxide-induced intestinal oxidative stress via the Pept1-p62-Nrf2 cascade [6].

Although the hydrolysis effects of ginger protease and other proteases have been compared in vitro studies, the variation of their oral absorption in vivo has not been explored. The diversity in amino acid composition and peptide sequences in collagen peptides may have an effect on the in vivo absorption [10]. The oral administration experiments showed that the GDCH was more efficiently absorbed by the gastrointestinal tract than hydrolysates obtained using pepsin, alkaline protease, and bromelain. This can be attributed to the following reasons. First, our results showed that GDCH was rich in dipeptides and tripeptides (42.19%), while previous studies have demonstrated that dipeptides and tripeptides can be absorbed intact in bioactive forms via oligopeptide transporter 1 (Pept1) on the intestinal brush border membrane [29]. Second, the independent transport systems of oligopeptides and free amino acids help to reduce the inhibition of absorption due to the competition between free amino acids for common absorption sites, which facilitates the functional effects of peptides as soon as possible. The existing research found that oligopeptides were absorbed and utilized more efficiently than free amino acids mainly because of their different transport systems [30,31]. Third, the molecular weight is an essential factor reflecting the degree of protein hydrolysis and is closely correlated with the efficiency of absorption and utilization in the body [10,32]. Thus, the high content of low-MW oligopeptides in the GDCH might contribute to the high bioavailability.

The conventional view is that protein must be degraded into amino acids before being absorbed and transported by small intestinal mucosal cells. However, recent studies have revealed that orally administered collagen hydrolysates are not entirely degraded to free amino acids and can be partially absorbed in the digestive tract as dipeptides and tripeptides in intact forms [11,33]. In this study, the results showed that GDCH was not entirely degraded to free amino acids and can be partially absorbed as dipeptides and tripeptides, such as Pro-Hyp, Gly-Pro-Hyp, and X-Hyp-Gly tripeptides. Interestingly, the Pro-Hyp and Pro-Hyp-Gly, not detected in GDCH, were identified in plasma after oral administration of the GDCH. It has been reported that Gly-Pro-Hyp-Gly can be partly absorbed as Pro-Hyp and Pro-Hyp-Gly by the gastrointestinal digestion [18]. Meanwhile, the Gly-pro-hyp can be degraded to Pro-hyp by the gastrointestinal tract [32]. Notably, several Hyp-containing peptides were identified in the blood, including Pro-Hyp, Hyp-Gly, and X-Hyp-Gly tripeptides, which were shown to be highly resistant to the degradation action of peptidases in plasma [11,18,34,35,36]. Moreover, studies have shown that these dipeptides and tripeptides were absorbed by the digestive tract after oral hydrolysates being transported to bloodstream and peripheral tissues [32,37,38] and excreted in the urine [39], suggesting the significant health benefits of GDCH in vivo.

The GDCH was rich in Gly-Pro-Y, Z-Pro-Gly, X-Hyp-Gly, Hyp-Gly, and Gly-Pro-Hyp-Gly. However, neither Pro-Hyp nor Gly-Pro-Hyp was identified, indicating that ginger protease cannot cleave Gly-Pro and Hyp-Gly bonds, which is in agreement with previous studies [9,18]. Furthermore, recent studies have reported that Gly-Pro-Y tripeptides (5%; *w/w*) were produced simultaneously with X-Hyp-Gly tripeptides up to 2.5% (*w/w*) using ginger protease. Furthermore, recent research has revealed that these specific active collagen oligopeptides exhibit a wide variety of physiological activities, including angiotensin-converting enzyme inhibitory activity [40], stimulation of the growth of skin fibroblasts [41], promotion of osteoblast differentiation [5], and prevention of intestinal oxidative stress [6]. Moreover, ginger protease’s unique substrate specificity for Hyp at the P_2_ position allows for the efficient production of X-Hyp-Gly-type tripeptides, which are basically undetectable in commercially available collagen peptides [5,18]. In addition, the Hyp-containing cyclic dipeptides, obtained from X-Hyp-Gly tripeptides by heating, were more efficiently absorbed into the blood [42]. Therefore, hydrolysate obtained by ginger protease can be considered as a potential source of bioactive peptides with a wide range of bioactivities and further studies are needed to explore its biological activity.

The hydrolysis site of ginger protease was determined based on the amino acids at the P_2_ position from 136 identified peptides. Meanwhile, the amino acid sequence of collagen characterized by repeating Gly-Xaa-Yaa triplet [43] can be observed in the 136 identified peptides. Another novel finding was that ginger protease can recognize Ala at the P_2_ position. Because of its hydrolysis specificity, ginger protease can be a promising tool for protein sequencing and identifying stable structural domains in proteins.

In addition, short peptides have shown great potential in the field of biomedical sciences. The relatively short peptides (di-, tri-, and tetra-peptides) can self-assemble into ordered nanostructures, such as nanotubes and fibrillar gels, which can be applied to drug delivery, tissue engineering, diagnostics, biosensing, and drug development [44,45]. Cell penetrating peptides are short peptides (<30 amino acids long) that can be used as a carrier for therapeutic agents, proteins, and SiRNA because of the ability to penetrate cellular lipid bilayers [46,47,48]. Small amino acid-derivatives that bind nucleic acids provide support for their further development as gene delivery agents [49,50]. Therefore, considering the potential bioactivity peptides screened by Peptide Ranker, more in-depth research should be conducted in the future to comprehensively explore their applications.

## 4. Materials and Methods

### 4.1. Material

Fish skin gelatin from tilapia (*Oreochromis mossambicus*) was purchased from Shanghai Xinxi Biotechnology Co. (Shanghai, China). The analysis certificate of gelatin was shown in Appendix A. Three proteases (pepsin, alkaline protease, and bromelain) were provided by Nanning Pangbo Bioengineering Co. (Nanjing, China). Additionally, trifluoroacetic acid (TFA), acetonitrile (LC-MS grade), trichloroacetic acid (TCA), sodium tetraborate, β-mercaptoethanol, 2,2-diphenyl-1-picrylhydrazyl (DPPH), sodium dodecyl sulfate (SDS), o-pthaldehyde (OPA), and dithiothreitol (DTT) were obtained from Sigma-Aldrich (St Louis, MO, USA). Ginger rhizomes were purchased from a local supermarket. The synthesized peptides (purity ≥ 98%) were obtained from Nanjing Peptide Biotech (Nanjing, China). Other reagents were of analytical grade or better.

### 4.2. Animal Experiment Ethics

Male SD rats (250−300 g) were supplied by SPF Biotechnology. (Beijing, China). All animal studies were carried out in compliance with the Guidelines for Care and Use of Laboratory Animals of Beijing University of Chinese Medicine and approved by the Animal Ethics Committee of the Centre of Experimental Animal, Beijing, China. (approval Nos. BUCM-4-2021122801-4116).

### 4.3. Preparation of Ginger Protease

Ginger protease was extracted using a methodology previously described with slight modifications [51]. Briefly, ginger rhizomes were homogenized in 2 volumes (*w*/*v*) of chilled 20 mM phosphate buffer, pH 7.2, containing 10 mM cysteine and 5 mM EDTA. Then, ginger protease was precipitated by ammonium sulfate (20% to 60% saturation) and dialyzed (using dialysis tubing with a cutoff of 8000–14,000 Da) for 24 h. After dialysis, ginger protease was lyophilized and stored at −20 °C. Finally, the activity of ginger protease was determined at 280 nm by measuring TCA-soluble peptides released from casein [52].

### 4.4. Preparation of Gelatin Hydrolysates

Alkaline protease, pepsin, bromelain, and ginger protease were used to hydrolyze gelatin under optimal conditions as shown in Table 7. First, the fish skin gelatin was dissolved in deionized water to obtain a 50 g/L gelatin solution. Then, the pH was adjusted with 1 mol/L NaOH or 1 mol/L HCl. The reaction was terminated by inactivating the enzymes in a boiling water bath for 5 min. After that, the mixture was centrifuged at 8000× *g* for 15 min. Supernatants were lyophilized by freeze-drying. The hydrolysates (HP, HA, HB, and GDCH) resulted from the hydrolysis of fish skin gelatin by pepsin, alkaline protease, bromelain, and ginger protease.

### 4.5. Degree of Hydrolysis Assay

The degree of hydrolysis (DH) was analyzed by modifying the OPA method proposed previously by Nielsen et al. [53]. In brief, serial dilutions of a known concentration of serine were used to establish a standard curve. Results were calculated as follows: DH (%) = (h/h_tot_) × 100%, where h_tot_ is the total number of peptide bonds per protein equivalent and h is the number of hydrolyzed bonds.

### 4.6. DPPH Radical Scavenging Activity

The DPPH experiment was performed according to a previously described methodology with some modifications [54]. Briefly, 1 mL of 0.1 mM DPPH in ethanol was added to an equal sample volume. Then, the mixture was incubated in the dark for 30 min at room temperature and the absorbance was measured at 517 nm. Results were calculated as follows: DPPH radical scavenging activity (%) = [1− (A_sample_ − A_o_)/(A_control_ − A_o_)] × 100%, where A_o_ was the absorbance without DPPH and A_sample_ and A_control_ were the absorbance of sample with DPPH and control without sample.

### 4.7. Oral Administration of Gelatin and the Hydrolysates (HP, HA, HB, and GDCH) in Rats

Hyp, a characteristic amino acid of collagen, is often used to quantify the absorption and dynamic changes of collagen peptides [10,11]. Therefore, the bioavailability was indirectly evaluated by the Hyp bioavailability after oral administration in rats [12,13]. After one week of acclimatization, the rats were randomly divided into five groups (*n* = 6 animals/group): gelatin Group, HA Group, HB Group, HP Group, and GDCH Group. Equal doses (2.4 g/kg body weight) were dissolved in distilled water to be orally administered to rats. Blood samples were collected from the orbital venous plexus before (0 h) and 0.5, 1, 2, 4, and 8 h after administration. Plasma was prepared by centrifugation of the blood at 860× *g* for 5 min at 4 °C and stored at −80 °C until analysis. The total Hyp dynamics in plasma were determined using a hydroxyproline assay kit (Jiangsu Kaiji Biotechnology Co., Nanjing, China).

### 4.8. Molecular Weight Distribution

The molecular weight distribution of GDCH was determined by a TSKGel G2000 SWXL (5 μm, 7.8 × 300 mm, TOSOH). Briefly, the sample (10 μL aliquot) was eluted with the mobile phase (45% aqueous acetonitrile solution containing 0.1% trifluoroacetic acid, *v*/*v*) at a flow rate of 0.4 mL/min. The elution was performed at 220 nm and 30 °C. A molecular weight calibration curve was obtained from the following standards: glycine trimer (189 Da), Gly-Gly-Tyr-Arg (451 Da), bacitracin (1423 Da), aprotinin (6512 Da), and cytochrome C (12,384 Da). The analysis was performed using Prominence^TM^ GPC System (Shimadzu, Kyoto, Japan).

### 4.9. Identification of Dipeptides and Tripeptides in Plasma after Intake of GDCH

Plasma was collected after 1 h from the orbital venous plexus and centrifuged (860× *g*, 5 min). Then the sample was deproteinized by the addition of three volumes of 100% ethanol and the supernatant was centrifuged at 14,000× *g* for 10 min at 4 °C. Finally, after filtration by a 3 K ultrafiltration membrane, 6 μL was injected into the LC-MS/MS system.

### 4.10. Identification of Dipeptides and Tripeptides by LC-MS/MS

LC-MS/MS was used to identify dipeptides and tripeptides in the plasma and GDCH. The LC was performed with the Vanquish UHPLC system (Thermo Fisher Scientific, Waltham, MA, USA) with an ACQUITY UPLC HSS T3 column (Waters, MA, USA). LC conditions used in this study were previously described [32,33]: binary gradient elution was performed with eluents A (0.01% (*v*/*v*) TFA) and eluents B (acetonitrile). The gradient profile with the following proportions (*v*/*v*) of acetonitrile was applied (0 min, 0%), (4 min, 0%), (9 min, 25%), (9.01 min, 80%), (10 min, 80%), (13 min, 0%), and (17 min, 0%). The flow rate was 0.3 mL min^−1^. A Thermo Scientific Q Exactive Plus Orbitrap LC-MS/MS System (Thermo Fisher Scientific, Waltham, MA, USA) was used with electrospray ionization (ESI) in the positive. Full scan mass spectral data were acquired from *m/z* 50 to 750. Capillary temperature, 320 °C; aux gas heater temperature, 400 °C; and spray voltage, 3.5 kV.

### 4.11. Peptide Identification and Sequencing by Nano LC-ESI-MS/MS

The GDCH was subjected to nano-LC-ESI-Q-Orbitrap-MS/MS for the determination of peptides profiles. The analysis conditions were as follows: Pre-column: Acclaim PepMap RPLC C_18_ 300 μm × 5 mm (5μm, 100 Å; Thermo Scientific, San Jose, CA, USA); analytical column: Acclaim PepMap RPLC C_18_ 150 μm × 150 mm (1.9 μm, 100 Å; Thermo Scientific, San Jose, CA, USA); mobile phase: (A) 2% acetonitrile with 0.1% formic acid and (B) 80% acetonitrile with 0.1% formic acid. The gradient profile with the following proportions (*v/v*) of mobile phase B was applied (0 min, 0%), (2 min, 8%), (45 min, 28%), (55 min, 40%), (56 min, 95%), and (66 min, 95%). The flow rate was 600 nL/min. Q Exactive^TM^ mass spectrometer ((ThermoFisher Scientific, San Jose, CA, USA) was operated in data-dependent acquisition (DDA) mode. The MS parameters used were: resolution, 70,000; (AGC) target, 3 × 10^6^; maximum injection time, 40 ms; and scan range, 100–2000 *m/z*. The MS/MS parameters used were: resolution, 17,500; (AGC) target, 1 × 10^5^; maximum injection time, 60 ms; TopN, 20; and normalized collision energy (NCE), 27. The sample was repeated twice, and the peptides duplicated in the two results were considered valid data for analysis.

PEAKS studio version 10.6 (Bioinformatics Solutions Inc., Waterloo, ON, Canada) was used for data analysis. The search parameters were as follows: uniprot database, (www.uniprot.org, accessed on 10 July 2022); species, *Oreochromis mossambicus (tilapia)*; fixed modifications, Carbamidomethylation: + 57.0215; variable modifications, Met(oxidation): + 15.99, Carbamoylation: + 43.01, Pro(hydroxylation): + 15.99, peptide N-term(acetylation): + 42.01, Asn and Gln(deamidation)NQ: 0.98; enzyme, no enzyme; maximum missed cleavages, 3; peptide mass tolerance, 20 ppm; fragment mass tolerance, 0.05 Da; mass values, monoisotopic. The peptides identified met a false discovery rate (FDR) ≤1% and a score >20.

### 4.12. Screening of Bioactive Peptides Using Bioinformatics Analysis

A total of 136 peptides (Appendix A) were obtained from the GDCH, which were subsequently screened by the Peptide Ranker database (http://distilldeep.ucd.ie/PeptideRanker/, accessed on 10 July 2022). Peptides were selected as potential bioactive peptides based on Peptide Ranker score (>0.8). These peptides were then subjected to Innowagen Website (http://www.innovagen.com/proteomics-tools, accessed on 10 July 2022) and ToxinPred Website (http://crdd.osdd.ner/raghava/toxinpred, accessed on 10 July 2022) for analysis.

### 4.13. Statistical Analysis

All the experiments were carried out at least in triplicate. The data were presented as mean ± standard deviation (SD). Data were subjected to analysis of variance (ANOVA), and mean comparison was performed using Duncan’s multiple range tests (SPSS, Version 22.0, IBM Inc., Chicago, IL, USA). Significant differences were defined at *p* < 0.05. The area under the curve (AUC) was analyzed using software GraphPad version 8.4.0 (USACO Corporation, Tokyo, Japan).

## 5. Conclusions

This study compared fish skin hydrolysates using ginger protease with those produced by other enzymes (pepsin, alkaline protease, and bromelain) in vitro and in vivo. Ginger protease appeared to be the most effective one in hydrolyzing gelatin. Further oral administration experiments revealed that GDCH was not completely degraded to free amino acids and can be absorbed as dipeptides and tripeptides in intact forms. In addition, LC-MS/MS results identified the unique substrate specificity of ginger protease recognizing Pro and Hyp at the P_2_ position. In the future research, further evaluations are needed to confirm the biological activities of hydrolysates using ginger protease in food industries and the biomedical field.

## Figures and Tables

**Figure 1 molecules-27-05001-f001:**
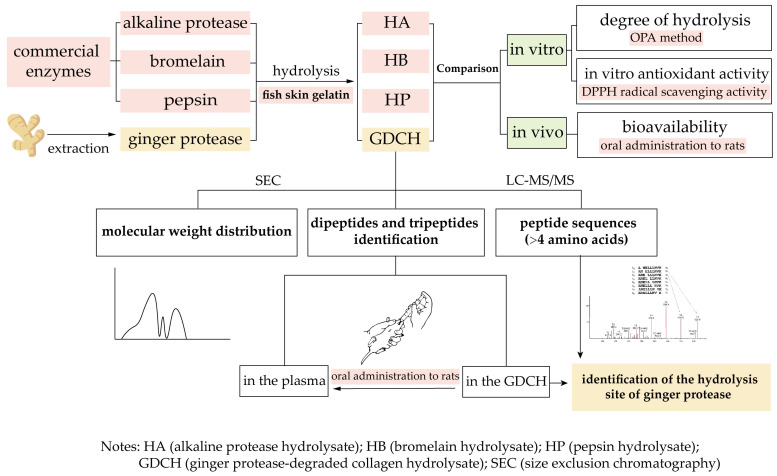
Flowchart of the study design.

**Figure 2 molecules-27-05001-f002:**
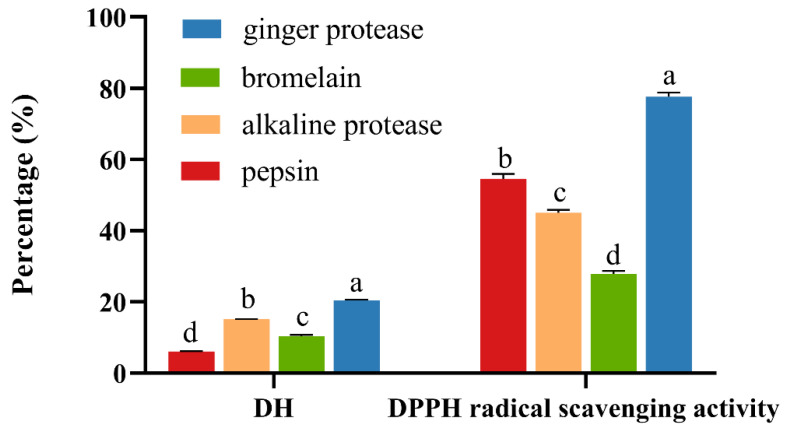
DH and DPPH radical scavenging activity of hydrolysates prepared by different proteases. Data were presented as mean ± SD (*n* = 3). Bar graphs followed by different letters indicate significant differences (*p* < 0.05) (ANOVA/Dunnet’s test). DH: degree of hydrolysis. DPPH: 2,2-diphenyl-1-picrylhydrazyl.

**Figure 3 molecules-27-05001-f003:**
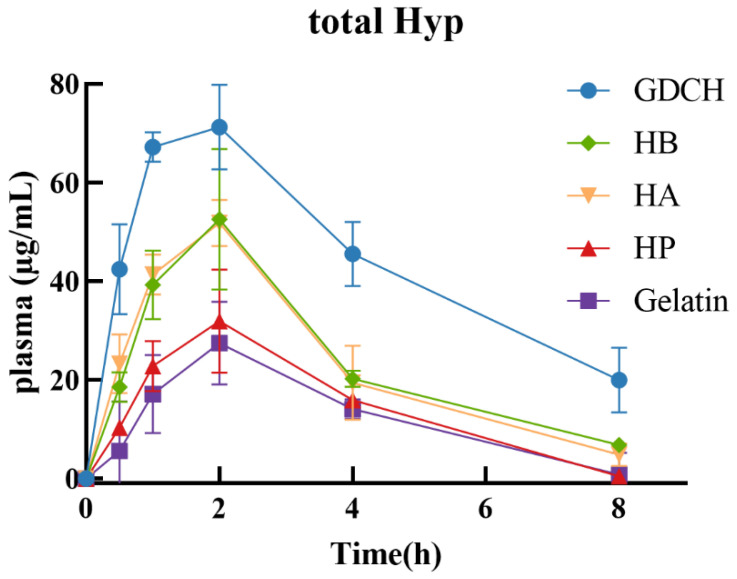
The concentration of total Hyp in rat plasma after oral ingestion (2.4 g/kg). Symbols: gelatin (■), HP (▲), HA (▼), HB (◆), GDCH (●). Data were shown as mean ± SD, *n* = 6. HA: alkaline protease hydrolysate; HP: pepsin hydrolysate; HB: bromelain hydrolysate; GDCH: ginger protease-degraded collagen hydrolysate.

**Table 1 molecules-27-05001-t001:** Pharmacokinetic parameters of the total Hyp in rat plasma after ingestion of gelatin and hydrolysates (HA, HB, HP, and GDCH).

Parameter	Gelatin	HP	HA	HB	GDCH
AUC_0–8 h_ (h µg mL^−1^)	100.65 ± 14.68 ^c^	118.72 ± 12.40 ^c^	188.29 ± 18.39 ^b^	191.86 ± 16.91 ^b^	355.01 ± 22.08 ^a^
C_max_ (µg mL^−1^)	27.44 ± 8.40 ^c^	31.89 ± 10.48 ^c^	51.82 ± 4.64 ^b^	52.54 ± 14.29 ^b^	71.24 ± 8.61 ^a^
T_max_ (h)	2.00 ± 0.00 ^a^	2.00 ± 0.00 ^a^	1.83 ± 0.41 ^a^	1.83 ± 0.41 ^a^	1.67 ± 0.52 ^a^

Results were shown as the mean ± SD from six independent samples (*n* = 6). Values followed by different superscript letters are significantly different at *p* < 0.05 (ANOVA/Dunnet’s test). HA: alkaline protease hydrolysate; HP: pepsin hydrolysate; HB: bromelain hydrolysate; GDCH: ginger protease-degraded collagen hydrolysate.

**Table 2 molecules-27-05001-t002:** The molecular weight distribution of the GDCH.

Molecular Weight Range/Da	Percentage of Peak Area/%
>10,000	0.11
5000–10,000	1.15
3000–5000	4.32
2000–3000	6.05
1000–2000	15.51
576–1000	26.45
189–576	42.19
<189	4.22

**Table 3 molecules-27-05001-t003:** The dipeptides and tripeptides in the GDCH.

Type		Peptide	t_R_/min	Formula	[M + H]^+^	Error/10^−6^	MS/MS Fragments (*m/z*)
Tripeptides	Gly-Pro-Y	Gly-Pro-Ala	3.2	C_10_H_17_N_3_O_4_	244.1291	0.410	70.0658, 127.0866, 155.0813, 187.1075
		Gly-Pro-Val	8.22	C_12_H_21_N_3_O_4_	272.1602	1.102	70.0658, 127.0866, 155.0813, 215.1388
		Gly-Pro-Gln	2.07	C_12_H_20_N_4_O_5_	301.1510	−1.328	70.0658, 127.0866, 155.0813, 244.1289
		Gly-Pro-Glu	3.75	C_12_H_19_N_3_O_6_	302.1342	1.655	70.0658, 127.0866, 155.0813, 245.1128
		Gly-Pro-Arg	1.93	C_13_H_24_N_6_O_4_	329.1929	0.608	70.0658, 127.0866, 175.1188
	Z-Pro-Gly	Ala-Pro-Gly	3.41	C_10_H_17_N_3_O_4_	244.1288	1.638	70.0658, 173.0910
		Leu-Pro-Gly	8.84	C_13_H_23_N_3_O_4_	286.1757	1.398	70.0658, 86.0969, 173.0919, 211.1438
	X-Hyp-Gly	Leu/Ile-Hyp-Gly	8.14	C_13_H_23_N_3_O_5_	302.1708	0.662	86.0605, 189.0868
		Phe-Hyp-Gly	7.95	C_13_H_23_N_3_O_5_	336.1551	0.892	86.0605, 120.0869, 189.0868
Dipeptides		Hyp-Gly	8.75	C_16_H_21_N_3_O_5_	189.0869	0.000	86.0606
Tetrapeptides		Gly-Pro-Hyp-Gly	1.31	C_7_H_12_N_2_O_4_	343.1615	−0.874	70.0658, 127.0866, 155.0813, 189.0867

**Table 4 molecules-27-05001-t004:** The dipeptides and tripeptides in rat plasma after oral ingestion of GDCH.

Peptide	t_R_/min	Formula	[M + H]^+^	Error/10^−6^	MS/MS Fragments (*m/z*)
Pro-Hyp	1.61	C_10_H_16_N_2_O_4_	229.1182	−0.873	70.0658, 132.0654
Gly-Pro-Hyp	2.82	C_12_H_19_N_3_O_5_	286.1397	−2.446	70.0658, 127.0866, 155.0813
Pro-Hyp-Gly	1.87	C_12_H_19_N_3_O_5_	286.1397	−1.048	70.0658, 86.0605, 189.0869
Leu/Ile-Hyp-Gly	8.14	C_13_H_23_N_3_O_5_	302.1708	0.662	86.0605, 189.0868
Phe-Hyp-Gly	7.95	C_13_H_23_N_3_O_5_	336.1551	0.892	86.0605, 120.0869, 189.0868

**Table 5 molecules-27-05001-t005:** The potential bioactive peptides obtained based on Peptide Ranker (>0.80) and ToxinPred Website.

No.	Peptides	Peptide Ranker Score	Formula	Mass/Da	Length	Toxicity
1	GPPGPPGPGP	0.948	828.413	C_38_ H_56_ N_10_ O_11_	10	no toxicity
2	GPXGPPGPGP	0.948	844.403	C_38_ H_56_ N_10_ O_12_	10	no toxicity
3	GPXGPXGPGP	0.948	860.3929	C_38_ H_56_ N_10_ O_13_	10	no toxicity
4	GPSGPXGPXGPXG	0.939	1117.4891	C_48_ H_71_ N_13_ O_18_	13	no toxicity
5	GPSGFXGPK	0.937	858.4186	C_39_ H_58_ N_10_ O_12_	9	no toxicity
6	PGXGGPXGPXG	0.935	933.4044	C_40_ H_59_ N_11_ O_15_	11	no toxicity
7	GPXGLXGPXGPA	0.924	1060.5042	C_47_ H_72_ N_12_ O_16_	12	no toxicity
8	GPXGLXGPPGPA	0.924	1044.5142	C_47_ H_72_ N_12_ O_15_	12	no toxicity
9	GPPVPGPIGP	0.900	886.4912	C_42_ H_66_ N_10_ O_11_	10	no toxicity
10	GPXGLXGPPGTP	0.893	1074.5247	C_48_ H_74_ N_12_ O_16_	12	no toxicity
11	SGPPVPGPIGP	0.890	973.5233	C_45_ H_71_ N_11_ O_13_	11	no toxicity
12	KAPDPFR	0.872	829.4446	C_38_ H_59_ N_11_ O_10_	7	no toxicity
13	GPPGPXGEEGKRGAR	0.869	1476.7384	C_61_ H_100_ N_22_ O_21_	15	no toxicity
14	GPXGPXGEEGKRGAR	0.869	1492.7284	C_61_ H_100_ N_22_ O_22_	15	no toxicity
15	GXAGNAGPXGPXGPA	0.867	1260.5587	C_53_ H_80_ N_16_ O_20_	15	no toxicity
16	G(+42.01)PAGNAGPXGPXGPA	0.867	1286.5792	C_55_ H_82_ N_16_ O_20_	15	no toxicity
17	GPXGPXGEEGKRGARGEXG	0.860	1848.8566	C_75_ H_120_ N_26_ O_29_	19	no toxicity
18	GPXGERGFXG	0.857	1001.4468	C_43_ H_63_ N_13_ O_15_	10	no toxicity
19	GXPGERGFXG	0.857	1001.4468	C_43_ H_63_ N_13_ O_15_	10	no toxicity
20	AGPXGPXGEKGSPG	0.857	1235.5684	C_52_ H_81_ N_15_ O_20_	14	no toxicity
21	GNAGPXGPXGPA	0.856	1019.4573	C_43_ H_65_ N_13_ O_16_	12	no toxicity
22	GPXGPXGEKGSPG	0.843	1164.5312	C_49_ H_76_ N_14_ O_19_	13	no toxicity
23	XAGNAGPXGPXGPA	0.843	1203.5372	C_51_ H_77_ N_15_ O_19_	14	no toxicity
24	GGXGERGAXGGRGFXG	0.836	1472.6609	C_60_ H_92_ N_22_ O_22_	16	no toxicity
25	AAGPXGPXGEKGSPG	0.826	1306.6055	C_55_ H_86_ N_16_ O_21_	15	no toxicity
26	NAGPXGPXGPA	0.826	962.4359	C_41_ H_62_ N_12_ O_15_	11	no toxicity
27	XGIGFPGPT	0.806	857.4233	C_40_ H_59_ N_9_ O_12_	9	no toxicity
28	GPPXGIVGPXGPA	0.801	1046.5298	C_47_ H_74_ N_12_ O_15_	12	no toxicity

X: hydroxyproline.

**Table 6 molecules-27-05001-t006:** Amino acids at P_2_ position.

Amino Acid	Total Number of Peptides	Percentage (%)
Pro	91	66.91
Hyp	20	14.71
Ala	13	9.56
Gly	5	3.68
Leu	2	1.47
Val	1	0.74
Phe	1	0.74
Met	1	0.74
Thr	1	0.74
Glu	1	0.74
total	136	100%

**Table 7 molecules-27-05001-t007:** The optimal conditions for the hydrolysis of fish skin gelatin by different enzymes.

Enzyme Type	Hydrolysis Conditions			Enzyme/Substrate (*w*/*w*)
	Optimum pH	Optimum Temperature (°C)	Hydrolysis Time (h)	
pepsin	2	37	4	0.05
alkaline protease	9	55	4	0.05
bromelain	6	55	4	0.05
ginger protease	6	55	4	0.05

## Data Availability

All data included in this study are available upon request by contact with the corresponding author.

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
