# Peer review of "Evaluating the Properties of Ginger Protease-Degraded Collagen Hydrolysate and Identifying the Cleavage Site of Ginger Protease by Using an Integrated Strategy and LC-MS Technology"

_molecules, 2022, doi:10.3390/molecules27155001_

Round 1

Reviewer 1 Report

This manuscript has an interesting idea aimed to compare fish skin gelatin hydrolysates using different proteases in vitro as well as in vivo. The authors observed that ginger protease exhibited the highest degree of hydrolysis as well as antioxidant activity. Furthermore, the in vivo absorption experiment of ginger hydrolysates indicated that different dipeptides and tripeptides were detected in the rat plasma. The topic of this paper is relevant, and of interest to the audience of this journal. However, there are some inquiries and comments which are mentioned below, therefore the manuscript is not acceptable for publication in this form. A native speaker or somebody good in English should revise the manuscript, in many sections meaning is lost due to poor English. See below corrections that can improve the manuscript.

Title:

The title needs to be improved and could be shortened, also the aim of work should be ameliorated.

Abstract:

The innovation of the manuscript should be clearly stated.

Abstract, lines 16: “the in vivo absorption and in vitro antioxidant properties” the order should be changed to first the in vitro then in vivo

Line 18: please change the word “Then”

Introduction

Please remember to write the abbreviation in full at the first time in which it is reported. Revise along the manuscript.  For example: Line 178 Ginger protease-degraded collagen hydrolysate replace to GDCH. Line 45, 283: Degree of hydrolysis (DH)

Results:

In all experiments, if the authors repeated the same experiments, please describe the number of experiments conducted.

Figure 1, how was the experimental design? How many repetitions of the experiment were carried out? How many analyses were performed? Statistical analysis?? I guess the results may vary within some range when they repeat the experiments.

Table 1: The Tmax of total Hyp after ingestion were 2.00±0.00 from six independent samples that means six measurements of six independent samples with the same results 2.00??? please clarify if it was biological replicates or three technical replicates?

The authors estimated total Hyp in rat plasma using seven replicates n=7 however, for pharmacokinetic parameters of the total Hyp in plasma they used only six replicates n=6 why ??

Figure 2 and table 1 why haven't the same samples order been used? negative control, gelatine?

Section 2.4: line 123: please move these sentences “indicating that ginger protease could not cleave Gly-Pro and Hyp-Gly bonds. The results agreed with previous studies [22]” to the discussion section

Section 2.5: The results are reported in a confusing way as it was written as result and discussion. please rewrite and better describe the results only.

Discussion:

Please, the authors could speculate the weaknesses and strengths of their study in Discussion section. What are the weaknesses and strengths of your study?

Many previous studies for examples Zheng et al., 2018 and Taga et al., 2016 clearly demonstrated the specific active collagen oligopeptides from gelatin hydrolysate prepared using ginger protease.  Although the authors declare that the peptide sequences in the GDCH was characterized for the first time? Please clarify and please highlight the novelty of this study. What was the impact of this study?

Reviewer 2 Report

This is an interesting manuscript as the authors have attempted to characterize the products of ginger protease activity including in vivo experiments.  

The following have to be addressed before the manuscript can be considered.

1. A few papers have reported on the use of ginger protease on fish skin gelatin. Can the authors highlight at the end of the introduction the novelty that their study brings to the area compared to previous studies.

2. Have the authors characterized the original gelatin used (viscosity, thermal, SDS page? Do they expect that the source of fish gelatin will impact on its amino acid composition and activity of the enzyme?

3. What was the original hydroxyproline content of the gelatin used to determine data in Figure 2.  The protocol used 'Equal doses (2.4 g/kg body weight)' may not reflect the equal content of hydroxyproline for good comparison.

Reviewer 3 Report

This work reports on the hydrolysis effectiveness of ginger protease compared to other hydrolytic means applied to fish skin gelatin and describes novel peptides identified from fish skin hydrolysates obtained by ginger protease treatment.

The work could be accepted for publication after the following revisions.

-English level should be checked and improved in all the text

-a mechanistic hypothesis should be reported on the antioxidant activity reported by the peptides obtained after the ginger protease hydrolysis of the fish skin gelatin

-more biomedical applications of short peptides as future perspectives should be mentioned including the ability shown by small amino acid-derivatives to bind nucleic acids citing the works with DOI: 10.1039/C6RA00294C and 10.1166/jbn.2017.2325

-Tab. 5: better explain which biological activity is expected from the Peptide Ranker score

-lines 391-393 not all abbreviations are explained. Check it! DPPH, DH...

-justify why in Tab 3 and 4 you refer to products of hydrolysis by ginger protease only and not to the others

-justify why in Tab S1 you refer to products of hydrolysis by ginger protease only and not pepsin, alkaline protease and bromelain. Are there already published results on their hydrolysis products starting from the same material (fish skin gelatin)?

Round 2

Reviewer 1 Report

The manuscript molecules-1825374 titled "Evaluating the properties of ginger protease degraded collagen hydrolysate and identifying the cleavage site of ginger protease by using an integrated strategy and LC MS technology” was improved after the authors consider the comments.  

However, there are some considerations:

Table 1:  The author changed the arrangement of HA, HB and HP without changing the values

-        Please avoid over citation as the authors add 17 new references, for example in the introduction 8 references was cited for only one sentence [10-17]. The same in discussion section five references were cited for one sentence [54-58], alternatively you can use one review article about biomedical application of short peptides or 2 recent research articles.

-        Please remove the abbreviation (BBM) and (CPPs) as they were used only one time in the manuscript.   

Reviewer 2 Report

The authors have addressed the comments.

Author Response

Thank you very much for your recognition to this manuscript.

Reviewer 3 Report

The article can be accepted in the current form

Author Response

(The authors gave the same response as above.)
